# Fatal human H3N8 influenza virus has a moderate pandemic risk

Valerie Le Sage[1,2]*, Michelle N. Vu[3], Maria A. Maltepes[4], Shengyang Wang[5], Brooke A. Snow[2], Grace A. Merrbach[2], Alexandra J. Benton[2], Kylie E. Zirckel[2], Sarah E. Petnuch[2], Carly N. Marble[2], Lora H. Rigatti[6], James C. Paulson[5], Elizabeth M. Drapeau[7], Anita K. McElroy[2,8], Scott E. Hensley[7], Louise H. Moncla[9], Seema S. Lakdawala [ID][3]*

1 Department of Microbiology and Molecular Genetics, University of Pittsburgh School of Medicine, Pittsburgh, Pennsylvania, United States of America, 2 Center for Vaccine Research, University of Pittsburgh School of Medicine, Pittsburgh, Pennsylvania, United States of America, 3 Department of Microbiology and Immunology, Emory University School of Medicine, Atlanta, Georgia, United States of America, 4 Department of Biology, School of Arts and Sciences, University of Pennsylvania, Philadelphia, Pennsylvania, United States of America, 5 Departments of Molecular Medicine and Immunology & Microbiology, The Scripps Research Institute, La Jolla, California, United States of America, 6 Division of Laboratory Animal Resources, University of Pittsburgh, Pittsburgh, Pennsylvania, United States of America, 7 Department of Microbiology, Perelman School of Medicine, University of Pennsylvania, Philadelphia, Pennsylvania, United States of America, 8 Division of Infectious Diseases, Department of Pediatrics, School of Medicine, University of Pittsburgh, Pittsburgh, Pennsylvania, United States of America, 9 Department of Pathobiology, School of Veterinary Medicine, University of Pennsylvania, Philadelphia, Pennsylvania, United States of America

* valerie.lesage@pitt.edu (VLS); seema.s.lakdawala@emory.edu (SSL)

## Abstract

In China, low pathogenic avian influenza (LPAI) H3N8 virus is widespread among chickens and has recently caused three zoonotic infections, with the last one in 2023 being fatal. Here we evaluated the relative pandemic risk of this 2023 zoonotic H3N8 influenza virus, utilizing our previously published decision tree. Serological analysis indicated that a large proportion of the human population does not have any cross-neutralizing antibodies against this H3N8 strain. LPAI H3N8 displayed a dual affinity for α2–3 and α2–6 sialic acids and replicated efficiently in human bronchial epithelial cells. Furthermore, we observed H3N8 transmission via direct contact but not aerosols to ferrets with pre-existing H3N2 immunity. Although pre-existing H3N2 immunity resulted in a shortened disease course in ferrets, it did not reduce disease severity or replication in the respiratory tract. This study suggests that this zoonotic H3N8 strain has moderate pandemic potential and emphasizes the continued need for avian influenza surveillance.

## Author summary

Low pathogenic avian influenza (LPAI) viruses circulate widely amongst birds and are a major public health concern for their ability to cross over to other

**Data availability statement:** The source data generated, analyzed and presented in Figs 2, 4, 5 and 6 of this study have been archived on FigShare (https://doi.org/10.6084/m9.figshare.c.7998790.v1). Plotting code for all phylogenetic analyses are available at https://github.com/moncla-lab/H3N8_pandemic_risk_paper, and full phylogenies from which these were subsetted are publicly viewable and interactive at https://nextstrain.org/groups/moncla-lab/h3nx/ha. All other segment trees are available and viewable via the dropdown menu under Dataset -> h3nx -> ha at https://nextstrain.org/groups/moncla-lab/h3nx/ha.

**Funding:** This project has been funded in part with Federal funds from the National Institute of Allergy and Infectious Diseases, National Institutes of Health, Department of Health and Human Services, under Contract No. 75N93021C00015 to SEH and SSL; VLS, BAS, AJB, KEZ, SEP and CNM were supported by NIH award (UC7AI180311) from the National Institute of Allergy and Infectious Diseases (NIAID), which supports the Operations of The University of Pittsburgh Regional Biocontainment Laboratory (RBL) within the Center for Vaccine Research (CVR); and Burroughs Wellcome CAMS 1013362.02 to AKM. LHM was funded by the National Institute of Allergy and Infectious Diseases at the National Institutes of Health (grant number R00-AI147029-05) and MAM is supported by funding from the Margaret Q. Landenberger Research Foundation. The funders had no role in study design, data collection and analysis, decision to publish, or preparation of the manuscript.

**Competing interests:** The authors have declared that no competing interests exist.

species, including humans. Here we characterize the pandemic potential of an H3N8 LPAI virus that caused a lethal human infection. While this strain was only able to transmit by direct contact, we found that it did exhibit some human adaptations, and pre-existing immunity did not reduce replication or pathogenesis, suggesting that it is a moderate pandemic risk and needs to be monitored given the potential public health threat.

## Introduction

Every year, seasonal influenza A viruses (IAV) cause respiratory infections in humans and impose major economic and health burdens on the human population. IAVs can infect a variety of different host species and because of this, new strains are constantly emerging through antigenic drift or reassortment, some with pandemic potential. An IAV pandemic can occur when, over time, the virus evolves traits that not only allow for cross-species transmission but also sustainable onward transmission within the new species. A measured public health response is crucial in the event of an emerging pandemic threat; we outlined a decision tree [1] to assess pandemic risk with the intention of guiding decision-making and preparedness efforts.

IAV are subtyped according to the two major surface glycoproteins, with 16 different hemagglutinin (HA) and 9 neuraminidase (NA) proteins. Many of the different combinations of HA and NA proteins are found in IAV strains from aquatic birds, making them a natural reservoir host [2–4]. Recently, highly pathogenic avian influenza (HPAI) outbreaks have been on the rise in poultry and wild birds. These infections are restricted to the H5 and H7 subtypes, result in high rates of mortality, and are a major public health concern. On the other hand, low pathogenic avian influenza (LPAI) viruses typically cause no or mild respiratory signs such as ocular and nasal discharge and swollen infraorbital sinuses. In wild bird populations, LPAI H3N8 is one of the most commonly found H3 subtypes and has also been found in mammalian hosts such as seals, horses, dogs and swine [5–9].

Since late 2021, LPAI virus subtype H3N8 has been detected in live poultry markets and farms in China [10] and has been associated with several spillover events into mammals. This lineage appears to have emerged from an influenza H9N2 virus that acquired a Eurasian avian lineage H3 and a North American avian lineage N8 [10,11]. A recent seroprevalence study of poultry workers from Hunan and Henan provinces in China indicated they were negative for H3N8 [11], suggesting that spillover frequency is quite low. However, in 2022 two different zoonotic H3N8 virus infections occurred in these regions in young boys exposed to poultry who became ill but subsequently recovered [11–13]. Additionally, a third H3N8 infection occurred in 2023 in a 56-year-old woman living in Guangdong Province, China, after exposure to poultry, which resulted in her death [14]. All three of these recent, human-infecting H3N8 strains descend from a group of avian H3Nx viruses circulating in Eurasia (Fig 1A) that frequently reassort (Fig 1B). In the 20 years prior to these 3 human infections, this HA has been associated with 5 different NA subtypes (N2, N3, N6, N8,

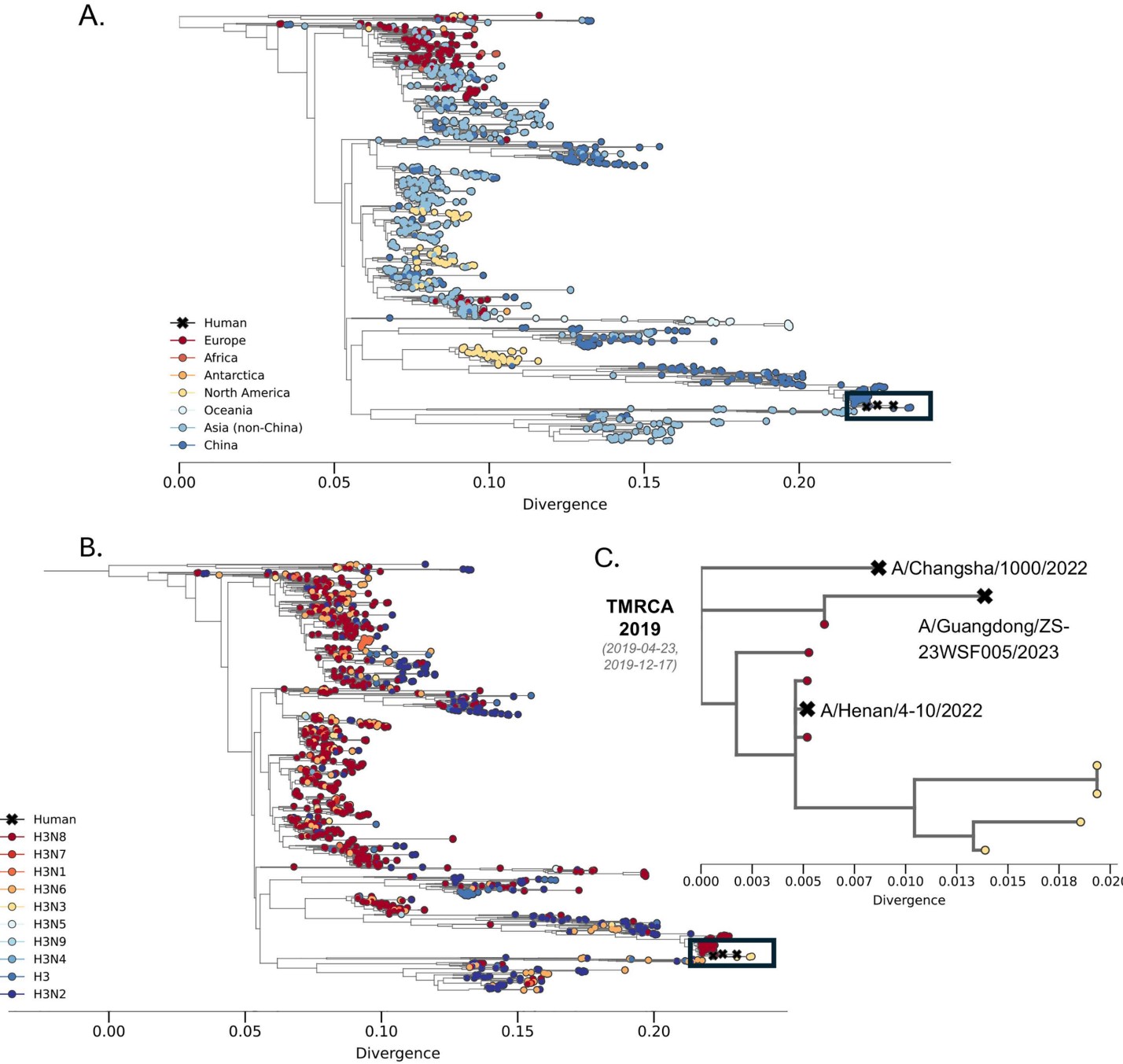

**Fig 1. Human infections with H3N8 nest within the diversity of frequently reassorting avian H3s circulating in Eurasia. A.** The HA phylogeny (N = 1,313) for the Eurasian avian lineage from which the three human H3N8 samples (A/Henan/4-10/2022, A/Guangdong/ZS-23SF005/2023, A/Chang-sha/1000/2022) descend from. Tips are colored by the geographic region of sampling origin. **B.** Same phylogeny as shown in A but colored by subtype where a change in color indicates a reassortment event of the NA gene occurred. The black box highlights where on the phylogeny the three human H3N8 isolates cluster, marked with Xs. **C.** An enlarged view of the human H3N8 samples from their most recent common ancestor. The estimated time to most common recent ancestor is annotated as the mean and the 95% confidence interval.

N9), including a switch from H3N2 to H3N8 prior to human spillover (Fig 1B). These human strains cluster together, with their HA gene tracing back to an inferred most recent common ancestor approximately four years prior (95% confidence interval April – December 2019) (Fig 1C).

Here, we characterized the pandemic potential of this third H3N8 strain (A/Guangdong/ZS023SF005/2023, herein referred to as A/GD/F005/23 H3N8) that resulted in fatality using our previously published decision tree [1]. We observed a lack of cross-reactive immunity against A/GD/F005/23 H3N8 in a representative population from the United States, which would allow rapid spread between individuals. Furthermore, the virus exhibited human adaptation characteristics by binding to both human-type and avian-type receptors as well as replicating efficiently in human bronchial epithelial cells. While transmission of A/GD/F005/23 H3N8 to ferret recipients with pre-existing H3N2 immunity was only detected via direct contact, replication and pathogenesis were similar in ferrets with and without pre-existing immunity, suggesting a moderate pandemic risk for A/GD/F005/23 H3N8.

## Results

### No detectable cross-reactive antibodies against H3N8 are present in individuals across different birth years

To begin to assess the pandemic potential of A/GD/F005/23 H3N8 (A/Guangdong/ZS023SF005/2023), we utilized a decision tree that we previously outlined in [1]. As population immunity is an important feature of assessing pandemic risk, we first determined whether sera samples collected from individuals in 2020 from the United States contained cross-reactive antibodies against A/GD/F005/23 H3N8. For comparison, we assessed immunity against the human seasonal viruses. Individuals from birth years ranging from 1940-2001 exhibited titers of neutralizing antibodies against a 2017 human seasonal H3N2 virus (A/Kansas/14/2017) and the A/California/07/2009 (H1N1pdm09) (Fig 2A). By contrast, only three individuals born in the 1950-1960s had low levels of neutralizing antibodies against A/GD/F005/23 H3N8. Sera from individuals pre- and post-vaccination with a quadrivalent flu vaccine were also analyzed for an increase in cross-reactive antibody titers. As expected, a rise in neutralizing antibodies from pre- to post-vaccination was observed with the positive control H1N1pdm09, but there was no increase in antibody titers against A/GD/F005/23 H3N8 (Fig 2B). Together, these data suggest that much of the United States population does not have protective cross-reactive antibodies to A/GD/F005/23 H3N8 through seasonal immunity or vaccination.

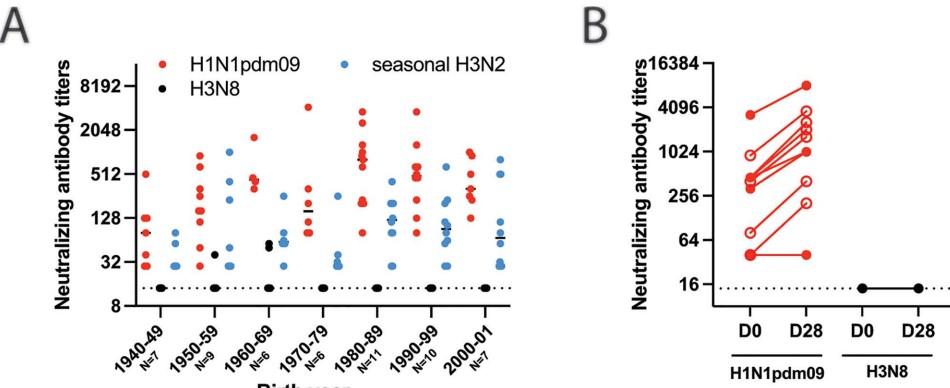

**Fig 2. Few individuals possess cross-reactive neutralizing antibodies against low pathogenic avian influenza A/GD/F005/23 H3N8 virus.**
**A.** Sera from the indicated number of individuals for each decade of birth were tested for antibodies to H1N1pdm09 (red), seasonal H3N2 (A/Kansas/14/2017, blue) and A/GD/F005/23 H3N8 (black) by neutralization assay. Data are presented as the mean values +/- standard deviation with each dot representing the neutralizing antibody titers of one individual. **B.** Plasma samples from ten individuals' pre-vaccination (D0) and post-vaccination (D28) with a quadrivalent flu vaccine were assessed for neutralizing antibodies. The slope of each line indicates the fold change in titer from pre- to post-vaccination, open circles identify those individuals with a greater than 4-fold rise. The dashed line indicates the limit of detection.

## H3N8 virus has affinity for α2,3 and α2,6 sialic acids

Whereby the HA protein of avian influenza viruses has a preference for α2–3 linked sialic acids (SA), the HA of human influenza viruses prefers α2–6 linked sialic acids [15–17]. Efficient airborne transmission of influenza virus has been correlated to a preference for α2–6 linked sialic acids [18–20]. We characterized the SA binding affinity of inactivated A/GD/F005/23 H3N8 virus using a glycan array of linear and N-linked sialosides (Fig 3A and 3B). The results indicated that the HA has dual specificity, binding to glycans capped with either α2–3 or α2–6 linked sialic acids (Fig 3C and 3D). Notably, however, the H3 HA bound preferentially to N-linked glycans with α2–6 sialic acids (Fig 3D). In contrast, a human seasonal H3N2 virus from 2021 (A/Darwin/06/2021) was used for comparison and specificity for binding only α2–6 sialic acids, and was observed to bind both the N-linked and linear sialosides (Fig 3E and 3F), consistent with other published reports [21]. These observations suggest that A/GD/F005/23 H3N8 is capable of binding to a large diversity of sialic acid moities.

## A/GD/F005/23 H3N8 replicates efficiently in human bronchiole epithelial cells

To further characterize the capacity of A/GD/F005/23 H3N8 to replicate in human airway epithelial cells, we examined virus replication in human bronchial epithelial (HBE) cells. The most recent influenza pandemic strain, H1N1pdm09, was used as a benchmark virus to compare phenotypes against the H3N8 virus. HBE cells are primary cells from patient lung explants that are differentiated and grown at an air liquid interface. Three distinct cultures of HBE cells were infected with either H1N1pdm09 or A/GD/F005/23 H3N8 virus and viral titers were determined at the indicated time points (Fig 4A). A/GD/F005/23 H3N8 was found to replicate in HBE cells similarly to H1N1pdm09 over time (Fig 4A), suggesting that A/GD/F005/23 H3N8 can infect and reproduce productively in human cells.

## *In vitro* characteristics of A/GD/F005/23 H3N8 are consistent with H1N1pdm09 phenotypes

Additional molecular features, such as pH of inactivation, NA activity, and persistence in the environment are all important factors for the success of influenza viruses in humans [22–25]. These *in vitro* characteristics of A/GD/F005/23 H3N8 were examined and compared to H1N1pdm09 strain as a benchmark for a successful recent human pandemic influenza virus. HA acid stability is an important transmissibility characteristic as a virus might be rendered noninfectious if HA undergoes a pH-dependent premature irreversible conformational change [25]. Avian influenza viruses have been reported to have a pH of fusion at >5.5, while human strains have a pH of fusion below 5.5 and closer to 5.0 [26–29]. To assess HA acid stability, we calculated the pH of inactivation for A/GD/F005/23 H3N8 and H1N1pdm09 (Fig 4B). Similar to prior reports, the 2009 H1N1pdm strain had a pH of inactivation of 4.9, while the pH of inactivation for A/GD/F005/23 H3N8 was 5.7. These results indicate that the acid stability of A/GD/F005/23 H3N8 maintains its avian phenotype, despite its ability to replicate efficiently in human airway epithelial cells (Fig 4A).

Environmental stability of influenza virus is also important for survival outside the host and subsequent transmission [30,31]. Our prior work has determined that human respiratory mucus can impact the persistence of influenza viruses at various relative humidity conditions, but this effect is specific for human seasonal viruses and not avian viruses [22,32]. Based on this previous research, environmental persistence of A/GD/F005/23 H3N8 virus was determined by mixing the virus with airway surface liquid (ASL) collected from three different HBE patient cell lines and stationary droplets were incubated at various relative humidities (RH). After a 2-hour incubation, the decay of A/GD/F005/23 H3N8 was calculated based on a time 0 recovery sample and compared to the decay of H1N1pdm09 at those same RH conditions (Fig 4C). Less than 2-fold decay of A/GD/F005/23 H3N8 was observed at different RHs, consistent with H1N1pdm09 decay phenotypes (Fig 4C), demonstrating that the environment stability of A/GD/F005/23 H3N8 is comparable to H1N1pdm09 in the presence of human respiratory mucus.

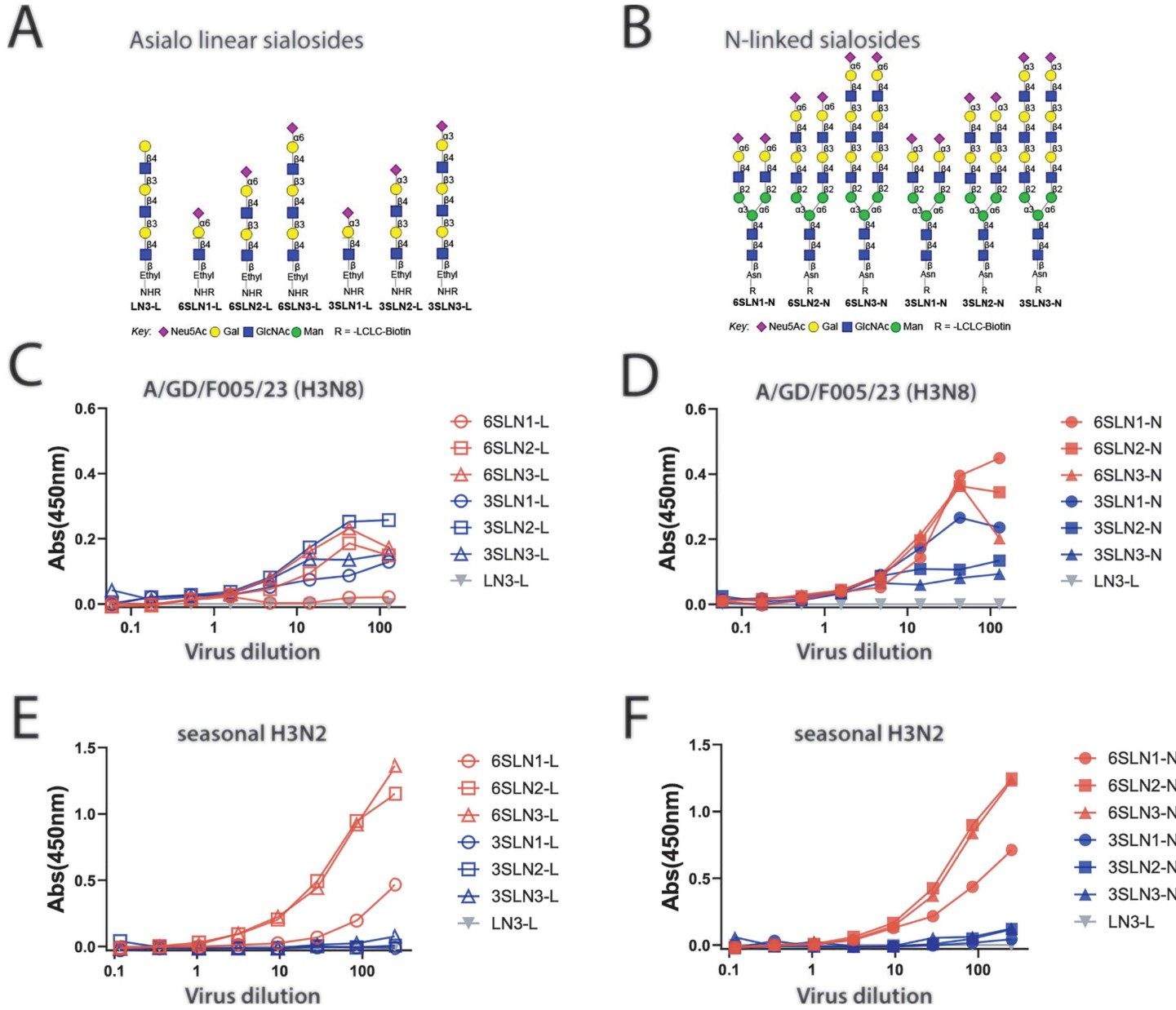

**Fig 3. Dual receptor binding specificity of A/GD/F005/23 H3N8.** Depiction of the biotinylated glycans with asialo and sialylated linear structures **(A)** or N-linked sialosides **(B)** having one to three LacNAc repeats. Binding of inactivated whole A/GD/F005/23 H3N8 virus to linear sialosides **(C)** or N-linked sialosides **(D)** was determined by an ELISA based format with α2-3 or α2-6 linked SA at an absorbance (Abs) of 450nm. As a control, binding of human seasonal A/Darwin/06/2021 H3N2 to linear sialosides **(E)** or N-linked sialosides **(F)** was performed to demonstrate the traditional binding of a seasonal H3N2 virus to α 2-6 linked SA. Control glycans with no SA (LN3-L, gray triangles) served as a negative control in C-F.

NA proteins on the virion surface promote the release of new virus progeny from infected cells by cleaving SA, and high NA activity has been implicated in promoting airborne transmission of influenza viruses [23,33,34]. To investigate NA activity, we utilized a fetuin based enzyme-linked lectin assay to compare A/GD/F005/23 H3N8 to H1N1pdm09. A/GD/F005/23 H3N8 NA activity was similar or moderately higher than H1N1pdm09 activity at all dilutions tested (Fig 4D), which would satisfy the need for an active NA to promote airborne transmission of the virus [23].

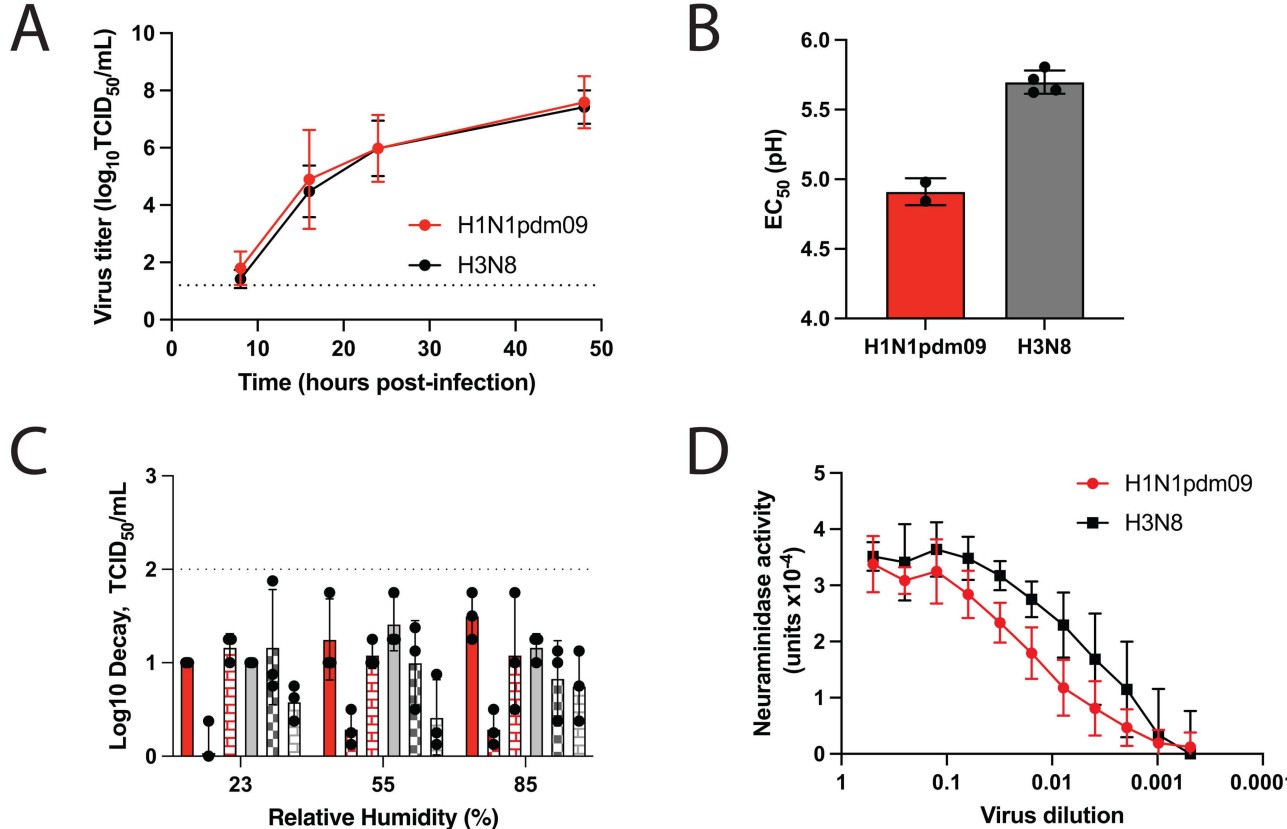

**Fig 4. In vitro characteristics of A/GD/F005/23 H3N8. A.** Human bronchial epithelial (HBE) cells were infected in triplicate with $10^4$ tissue culture infectious dose ($TCID_{50}$) of H1N1pdm09 (red) or A/GD/F005/23 H3N8 (black) and supernatant collected at the indicated time points. Virus was titered on Madin Darby canine kidney (MDCK) cells by $TCID_{50}$ assay and the data shown are the mean ± standard deviation of three patient HBE cell lines. **B.** H1N1pdm09 (red) or A/GD/F005/23 H3N8 (grey) viruses were incubated at a dilution of 1:100 in phosphate-buffered saline (PBS) solutions of decreasing pH increments for 1 hour at 37˚C, with experiments performed in triplicate. Remaining viable virus was titered by $TCID_{50}$ assay on MDCK cells and the half maximal effective concentration ($EC_{50}$) values at which 50% of virus was inactivated were plotted using regression analysis of the dose-response curve. The reported mean (±standard deviation) corresponds to biological replicates, each performed in triplicate. A paired t-test suggests a significant difference between these two viruses (p-value<0.005). **C.** Stocks of H1N1pdm09 (red bars) or A/GD/F005/23 H3N8 (grey bars) were each diluted 1:10 in ASL from three different HBE cell cultures. The virus in ASL suspension was used to make ten 1µL droplets in triplicate, which were incubated under three different relative humidity (RH) conditions for 2 hours. Infectious titers were then determined by $TCID_{50}$ assay on MDCK cells and presented as the average log decay from time 0. Each ASL is a separate bar, and the technical replicates are shown as individual points per bar; error bars represent ±standard deviation of the technical replicates. **D.** H1N1pdm09 (red) and A/GD/F005/23 H3N8 (black) NA activities were determined using virus diluted in PBS in an enzyme-linked lectin assay (ELLA) with fetuin as the substrate. Viruses were normalized for equal infectivity on MDCK cells and displayed data are representative of three independent ELLA experiments performed in duplicate. Results are displayed as the mean NA activity at $10^5$ $TCID_{50}$/mL (±standard deviation).

Taken together with our risk assessment triage tree [1], the results of these *in vitro* assays indicate that while A/GD/F005/23 H3N8 has 2 of the 3 molecular features thought to promote airborne transmission (environmental stability across different RHs and NA activity but not pH of inactivation). Importantly, the pH of inactivation is above 5.0 and studies in swine and human H3N2 viruses have suggested that a virus with pH of inactivation at 5.5 may still transmit with some efficiencies [27,35].

### A/GD/F005/23 H3N8 transmits via direct contact but not aerosols in ferrets with prior H3N2 immunity

By the age of 5, most humans have been exposed to and mounted an immune response against influenza virus [36], meaning that zoonotic disease is emerging in the context of population immunity. Thus, for pandemic risk assessment,

prior immunity is an important feature to consider when assessing transmission fitness to ensure that the pandemic risk for a given subclade is not overstated. To assess the transmission capacity of A/GD/F005/23 H3N8 in the context of immunity, we used our previously developed ferret model in which an infected donor ferret is exposed to a recipient ferret with pre-existing immunity [37]. For this study, recipient immunity was obtained from a prior infection with the same subtype of influenza virus as H3N8, the H3N2 A/Perth/16/2009 strain (hereafter referred to as H3N2-imm), 112 days prior to the transmission study with H3N8 (Fig 5A and 5B). Prior immunity to seasonal H3N2 virus did not provide any cross-neutralizing antibodies against the avian H3N8 virus (Fig 5C and 5D).

To assess airborne transmission, H3N2-imm recipients were separated from an A/GD/F005/23 H3N8 infected donor by a perforated divider within a cage with directional airflow for 2 days. While all three A/GD/F005/23 H3N8 infected donors

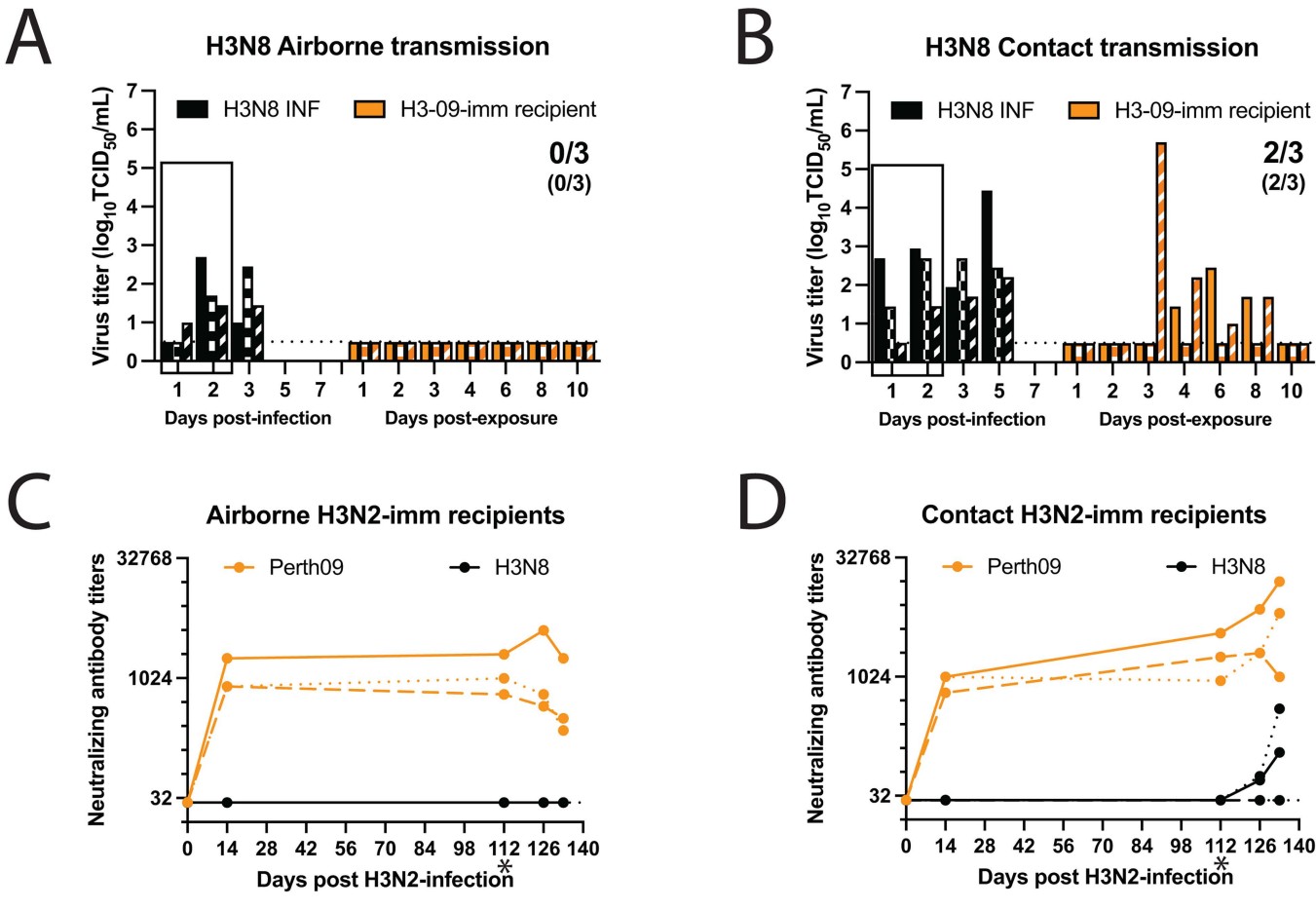

**Fig 5. A/GD/F005/23 H3N8 transmits via direct contact but not aerosols in ferrets with prior H3N2 immunity.** Six ferrets were infected with $10^6$ $TCID_{50}$ of A/Perth/16/2009 to act as recipients in airborne and contact transmission studies 112 days later. Twenty-four hours after three donor ferrets were intranasally infected with $10^6$ $TCID_{50}$ of A/GD/F005/23 H3N8, donors were exposed to recipient ferrets having pre-existing H3N2 immunity (H3N2-imm) by airborne transmission **(A)** or direct contact transmission **(B)**. The box indicates the time in which the donors were exposing the H3N2-imm recipients. Nasal washes were collected from all ferrets on the indicated days and each bar represents the virus titer from an individual ferret as determined by $TCID_{50}$ assay on MDCK cells. Numbers indicate the recipients infected while numbers in parentheses indicate recipients that seroconverted. Limit of detection is indicated by the dashed line. Neutralizing antibody titers against (H3N2; orange line) or A/GD/F005/23 H3N8 (black line) from the airborne **(C)** or direct contact **(D)** recipient ferrets. At the indicated days post-H3N2 infection, serum was collected, and neutralizing antibodies were determined by microneutralization assay. Each line represents an individual animal. The '*' denoted the start of the transmission study. Limit of detection is indicated by the dashed line.

shed virus in nasal washes, none of the three H3N2-imm recipients were found to shed A/GD/F005/23 H3N8 in nasal washes (Fig 5A) or seroconvert (Fig 5C) under these conditions. To measure direct contact transmission of A/GD/F005/23 H3N8, H3N2-imm recipients were co-housed with the H3N8 infected donors for 2 days without separation. Nasal washes were performed to assess viral shedding and revealed that transmission of A/GD/F005/23 H3N8 occurred in two of three H3N2-imm recipients (Fig 5B). Sera samples taken from direct contact recipients also confirmed this result (Fig 5D). We also noted an increase in H3N2 neutralizing antibodies in both A/GD/F005/23 H3N8 positive direct contact recipients likely due to a cross-reactive memory response stimulated by H3N8 infection (Fig 5D). Ferrets either intranasally infected with $10^6$ TCID$_{50}$ A/GD/F005/23 H3N8 inoculum or infected via exposure displayed only mild clinical signs such as mild nasal discharge and reduced activity on no more than two consecutive days. These data indicate that A/GD/F005/23 H3N8 can transmit via direct contact but not aerosols to animals with prior H3N2 immunity.

### Human seasonal H3N2 immunity reduces disease duration of A/GD/F005/23 H3N8

 The impact of pre-existing H3N2 immunity on A/GD/F005/23 H3N8 virus replication and disease severity was assessed by comparing intranasally A/GD/F005/23 H3N8 infected naive and H3N2-imm donor ferrets. Organs from the respiratory tract were collected on days 3 or 5 post-infection and analyzed for infectious virus as well as lung injury. Infectious viral titers in the lungs, trachea, and soft palate were not significantly different between the two groups of ferrets on day 3 post-infection (Fig 6A) but were greatly reduced in H3N2-imm ferrets on day 5 (Fig 6B). Interestingly, viral loads in the nasal turbinates were significantly reduced in pre-immune ferrets on both days 3 and 5 post-infection. A/GD/F005/23 H3N8 virus replication was measured in nasal washes and appeared similar between the two groups of ferrets on days 1–3 post-infection, with the only statistical significance difference occurring on day 5 post-infection in H3N2-imm ferrets having no detectable virus (Fig 6C). Lung damage on day 3 post-infection was similar between both groups of animals, whereas lung pathology scores were significantly higher in naïve ferrets as compared to H3N2-imm ferrets on day 5 post-infection (Figs 6D and S1 Fig). Together, these results suggest that seasonal H3N2 immunity may shorten the disease course but has only a modest impact on initial A/GD/F005/23 H3N8 replication and lung disease.

## Discussion

The recent fatal human H3N8 virus infection in China prompted interest in this LPAI strain. Here, we show that this H3N8 strain, A/Guangdong/ZS023SF005/2023, is a moderate pandemic risk according to our previously described decision tree [1], which utilizes a combination of *in vitro* and *in vivo* assessments. A/GD/F005/23 H3N8 is antigenically novel within a representative US human population. *In vitro* characterization revealed dual binding specificity of H3N8 to α2–3 and α2–6 SA, as well as similar environmental decay across different RHs and NA activity to the H1N1pdm09 strain. These molecular features are consistent with epidemiologically successful influenza viruses, but the lack of airborne transmission to ferrets with human seasonal H3N2 immunity reduces the risk to moderate compared to other circulating zoonotic strains. It is feasible that the virus could transmit via the air to immunologically naïve ferrets as has been suggested for other recent H3N8 strains [38], but this may not accurately reflect what could occur in a human population with varying levels of immunity. Interestingly, contact transmission to two of three H3N2-imm ferrets was observed suggesting that the virus can spread and infect ferrets with prior immunity. Pathology studies suggest that prior human seasonal H3N2 immunity reduces the course of the disease but did not alter the viral load in all respiratory tissues tested nor lung pathology during the initial phase of infection.

   Prior immunity to human seasonal H3N2 viruses is high within the human population, given the seasonal circulation since 1968. However, there is considerable evolution of seasonal H3N2 viruses and immunity to one may be different from others, thus our choice of the 2009 H3N2 strain may only represent those imprinted after 2010. Repeated exposure to drifted H3 strains could also impact the immune landscape and is not represented in our single immune imprinted animals. In addition, imprinting or exposure to more recent H3N2 viruses may produce a different antibody response than the

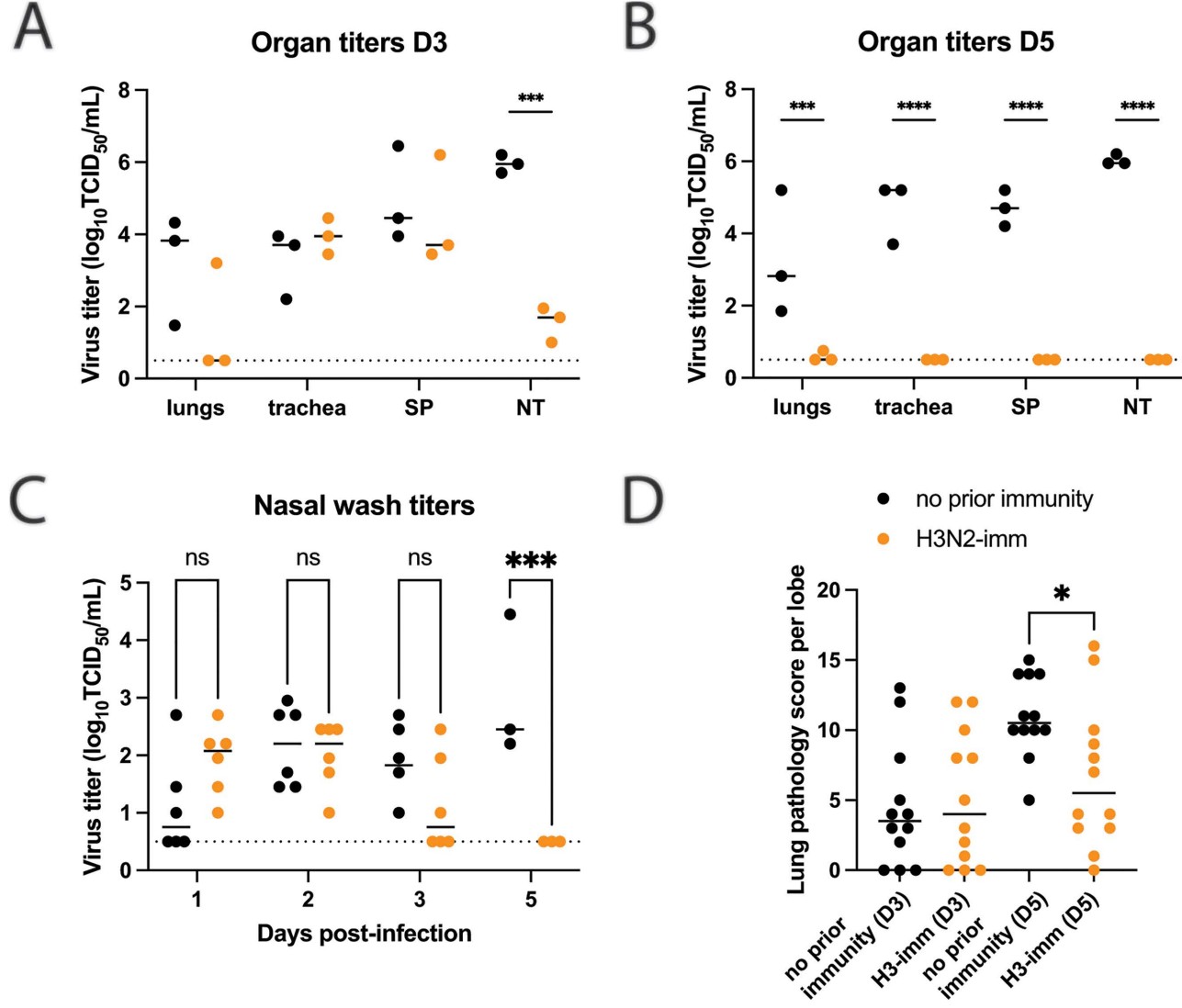

**Fig 6. Pre-existing H3N2 immunity leads to faster H3N8 clearance.** Ferrets with no prior immunity (black) and those previously infected with A/ Perth/16/2009 112 days prior (H3N2-imm, orange) were intranasally infected with $10^6$ $TCID_{50}$ of A/GD/F005/23 H3N8. Respiratory tissues from H3N2-imm ferrets or those with no prior immunity were collected on days 3 **(A)** and 5 **(B)** post-infection with A/GD/F005/23 H3N8. SP-soft palate, NT-nasal turbinates. The median±standard deviation of viral titers is shown with each circle representing the titer from an individual ferret. Two-way ANOVA analysis was used to detect statistically significant differences (***p-value=0.0010 and ****p-value<0.0001). **C.** Nasal washes were collected on the indicated days post-infection and titered on MDCK cells by $TCID_{50}$ assay with each circle representing the titer from an individual ferret. The median virus titer is indicated by the line. Statistical significance was determined using two-way ANOVA (***p-value=0.0007). **D.** Lung pathology scores of ferrets with no prior immunity (N=3) or H3N2-imm (N=3) were determined on days 3 or 5 post- A/GD/F005/23 H3N8 infection with four lobes being scored for each animal. Each circle represents the lung pathology score per lobe and the line represents the median score. Statistical significance was determined using an unpaired t-test (p-value=0.0187).

2009 H3N2 strain. It is still unclear what the best immune imprints are for risk assessment, and the impact of more recent strains or repeat seasonal infections should be considered in future risk assessment studies.

A major determinant of host range restriction is thought to be receptor binding specificity and is a known barrier to air- borne transmission of avian viruses in the ferret model [3,39]. Surprisingly, the human derived avian-origin A/GD/F005/23 H3N8 strain binds both avian-type and human-type receptors with preference for human-type receptors. Two other

isolates from human infections by avian-origin H3N8 IAV strains in 2022 also exhibited dual receptor specificity, although with a stronger preference for the avian-type receptor [11]. Analysis of representative chicken-origin H3N8 IAV strains isolated in 2022 also exhibited dual receptor binding specificity, indicating that the dual specificity was not a result of further adaptation within the human hosts [11]. Another study suggested that the human H3N8 isolates had a greater capacity for replication in normal human bronchial epithelial cells than chicken isolates [38]. Because our assays differ somewhat from those used in the earlier study, it is not possible to say that the preference for human-type receptors that we observed in the H3N8 strain represents a significant change from the cases reported in 2022. It is important to note that in the 1918 and 2009 H1N1 human pandemics some early isolates exhibited dual receptor specificity resulting from a single mutation from the avian virus progenitor [40–42]. Thus, we consider the existing dual receptor specificity of the H3 HA of the H3N8 virus already sufficiently adapted to mediate human-to-human transmission.

Of the three A/GD/F005/23 H3N8 zoonotic infections since 2022, two strains (A/Henan/4–10/2022 [or HN/4–10] and A/Changsha/1000/2022) have been previously characterized by Sun and colleagues [38]. Of these two human H3N8 isolates, only HN/4–10 was able to transmit efficiently via the air, yet both viruses transmitted by direct contact to ferrets with no prior immunity. The A/GD/F005/23 H3N8 strain characterized here contains 67 and 97 residues that differentiate it from the Henan and Changsha human H3N8 strains, respectively, scattered throughout the genome (all amino acid differences listed in S1 Table). These viruses have diverged for ~4 years and harbor amino acid changes in every gene segment, making them genetically distinct from each other. The high number of amino acid differences could partially explain some differences in observed transmissibility and infectivity between strains. HN/4–10 contains the PB2 E627K mutation, which is not only essential for respiratory droplet transmission in ferrets [3] but is also important for mammalian adaptation [43,44]. The mild disease observed in A/GD/F005/23 H3N8 infected ferrets may be the result of the E at position 627, in stark contrast to the more severe signs (lethargy, loss of appetite and ruffled fur) observed by Sun and colleagues [38]. The observed differences in airborne transmission between these other H3N8 strains and the one tested here may stem not only from sequence differences, but also from pre-existing H3N2 immunity, which was not assessed in the other publication. Regardless, taken together, the published data and our results suggest efficient direct contact transmission and limited airborne transmission, indicating a moderate risk of this virus in the absence of additional mammalian adaptations.

The risk of an emergent IAV is increased when there is a lack of existing human immunity. Although a serosurvey of farm poultry workers in China would indicate that there is a low prevalence of H3N8 virus exposure in this population [11], these data also demonstrate a lack of cross-reactive antibodies in people at this high-risk human-animal interface. Similarly, our representative US population had virtually no cross-reactive neutralizing antibodies above the limit of detection, except for three individuals that were born in the 1950s and 1960s. Furthermore, neither seasonal H3N2 vaccination in China [38] nor in the US (Fig 2B) were able to generate antibodies that cross-reacted against H3N8 IAV strains. However, non-neutralizing immunity from prior H3N2 infection could protect from clinical disease, viral load, and susceptibility to infection. Therefore, the overall risk of current zoonotic H3N8 is moderate to low in the face of prior seasonal H3N2 immunity.

Continued H3N8 spillover events into marine mammals have been documented, including seals [9], and provide an opportunity for the virus to evolve and escape the protection conferred by prior H3 immunity. The three spillover events of H3N8 were a result of exposure to live poultry, reinforcing the importance of surveillance of the natural reservoirs in migratory waterbirds for avian influenza virus, which have the potential to intersect and result in incursions of H3N8 influenza virus into other species.

## Materials and methods

### Ethics statement

All ferret experiments were conducted in the University of Pittsburgh's BSL2 facility in compliance with the guidelines of the Institutional Animal Care and Use Committee (approved protocol 22061230). Isoflurane was used to sedate animals

for all nasal washes and survival blood draws, as directed by approved methods. For terminal procedures, animals received ketamine and xylazine for sedation, followed by euthanasia solution administered via cardiac injection.

## Phylogenetic reconstruction

The Eurasian avian lineage tree from which the three human H3N8 samples descend from was subsetted from the global phylogeny of H3Nx evolution built and maintained by Maria Maltepes and the Moncla lab using the Nextstrain pipeline [45] (full trees available at nextstrain.org/moncla-group/h3nx/ha). Sequences were downsampled to 30 sequences per year, country, host, and subtype and then aligned with MAAFT [46]. A maximum likelihood tree was inferred using IQTree [47], and a time-resolved phylogeny and inference of subtype and host at internal nodes were inferred using TreeTime [48]. Trees were plotted using Baltic v.0.3.0 (ref: https://github.com/evogytis/baltic). All plotting, tree manipulation code, and GISAID accessions used are publicly available on the Moncla lab Github workspace at https://github.com/moncla-lab/H3N8_pandemic_risk_paper. A list of all accession numbers used in these analyses, along with corresponding GISAID acknowledgments, are included in S2 Table.

## Cells and viruses

The American Type Culture Collection (ATCC) provided Madin-Darby canine kidney (MDCK) and 293T cells, which were cultured in Eagle's minimal essential medium (MEM) containing 10% fetal bovine serum, 2mM penicillin/streptomycin and 2 mM L-glutamine. Primary human bronchiole epithelial (HBE) cell cultures were derived from human lung tissue and maintained at an air-liquid interface as outlined in the protocol approved by the institutional review board at the University of Pittsburgh [49]. All cells were cultured at 37$^\circ$C and 5% CO2. Reverse genetic derived strains of A/California/07/2009 and A/Perth/16/2009 were a generous gift from Dr. Jesse Bloom (Fred Hutch Cancer Research Center, Seattle) and previously rescued as below. A/Darwin/06/2021 (FR-1837) and A/Kansas/14/2017 (FR-1666) were obtained from Influenza Reagents Resources.

## Rescue of virus from reverse genetics plasmids

Reverse genetics plasmids expressing A/Guangdong/ZS023SF005/2023 (H3N8) were synthesized based on sequence deposited in GISAID (Accession number EPI2508604). Noncoding regions (NCRs) were present for everything except PA, NA, and NP. To fill in the NCRs for PA, NA, and NP, a set of pan-avian H3Nx trees was assembled. Using the phylogeny, ~30–60 sequences on the tree that were closest to the human strain were re-aligned with the human strain, and the missing regions were compared. For PA, NA, and NP, all the nearest neighbor sequences that had complete NCRs had identical NCRs, so that sequence was used to fill in the missing gaps in the human strain. The eight reverse genetics plasmids were transfected into 293T cells using Lipofectamine 2000 in Opti-MEM complete media. Twenty-four hours later, MDCK cells in Opti-MEM complete media containing 0.5ug/ml tosyl phenylalanyl chloromethyl ketone (TPCK)-treated trypsin were overlay into the 293T cells and cytopathic effect (CPE) was monitored over 2 days. The supernatant was collected and passaged onto confluent MDCK cells to expand the virus stock (cell passage 1, cP1). A second cell passage (cP2) stock was generated on MDCK cells and used for all subsequent experiments. Plasmids were sequence verified prior to rescue.

## Virus titration

Titrations were performed by tissue culture dose 50 (TCID$_{50}$) using confluent 96 well plates of MDCK cells in Eagle's MEM containing Anti-Anti, L-glutamine, and 0.5ug/ml TPCK-treated trypsin. The sample was added to the first row of wells in quadruplicate and tenfold serial dilutions were performed. Cells were scored for CPE after a 96-hour incubation period. Virus titers were calculated using Reed and Muench method [50] and expressed as log$_{10}$ TCID$_{50}$/mL.

## Human subject statement

The human serum samples, used in Fig 2A, were collected from healthy adult donors in Pittsburgh, Pennsylvania who provided written informed consent for their samples to be used in infectious disease research under the University of Pittsburgh Institutional Review Board approved protocol STUDY20030228. Participants self-reported their age, sex, race, ethnicity, residential zip code, history of travel and immunizations. In Fig 2B, plasma samples were collected from healthy adult donors vaccinated with FluLaval Quadrivalent in Philadelphia, Pennsylvania under the University of Pennsylvania Institutional Review Board approved study 849398. The University of Pittsburgh Institutional Review Board approved protocol STUDY19100326 for collection of deidentified patient lungs to generate human bronchiole epithelial cell cultures.

## Microneutralization assay

To inactivate non-specific inhibitors, human and ferret sera or plasma samples (one part) were treated with three parts receptor-destroying enzyme (RDE) II (Hardy Diagnostics) at 37 °C overnight. RDE was inactivated by incubation at 56 °C for 30 minutes. Two-fold serial dilutions of RDE-treated samples were incubated with $10^{3.3}$ $TCID_{50}$ of influenza virus for 1 hour at room temperature. Virus:serum/plasma mixture was added to confluent MDCK cells in a 96-well plate with Eagle's MEM containing Anti-Anti, L-glutamine and 0.5ug/ml TPCK-treated trypsin. After 4 days, CPE was determined, and the neutralizing antibody titer was expressed as the reciprocal of the highest dilution of serum/plasma required to completely neutralize the infectivity of each virus on MDCK cells. The concentration of antibody required to neutralize 100 TCID50 of virus was calculated based on the neutralizing titer dilution divided by the initial dilution factor, multiplied by the antibody concentration.

## Replication in human bronchiole epithelial cell cultures

The apical surface of the HBE cell cultures was washed with phosphate-buffered saline (PBS) to remove airway surface liquid (ASL) and then each transwell was infected with $10^3$ $TCID_{50}$ of virus in 100 µL HBE growth medium. Residual inoculum was collected after 1 hour incubation and cells were washed with PBS three times. Released virus particles were collected at the indicated time points by adding 150 µl of HBE growth medium to the apical side for 10 minutes. The $TCID_{50}$ endpoint method [50] was used to determine viral replication.

## Glycan ELISA

Streptavidin-coated 384-well plates (Pierce) were washed with PBS three times followed by the addition of 50 µL a biotinylated glycan in PBS and incubated at 4 °C overnight. Glycans were synthesized by authors J.P and S.W as described in [51]. The plates were rinsed with 0.05% Tween 20 in PBS (PBS-T) five times to remove excess glycans. Each well was then blocked with 100 µL of 1% bovine serum albumin (BSA) in PBS containing 0.6 µM desthiobiotin at room temperature for 2 h. Plates were then washed with PBS-T five times and used without further processing. The virus sample was subjected to 3-fold serial dilutions. Fifty microlitre of each sample was then transferred to wells of glycan-coated plates and incubated at 4 °C overnight. The wells were rinsed with PBS-T five times and incubated with 50 µL of snowdrop lectin-HRP conjugate at room temperature for 2 h. After washing with PBS-T, the wells were filled with 50 µL of the 3,3′,5,5′ tetramethylbenzidine (TMB) peroxidase substrate until it gave good color titers. The reactions were quenched with 50 µL of 2M sulfuric acid. The absorbance at 450 nm was detected using a microplate reader.

## pH of inactivation assay

Influenza virus was incubated at a ratio of 1:100 (10 µL in 990 µL) in pH-adjusted PBS that was titrated from 7.5 to 3.0 at roughly 0.5 pH increments using concentrated HCl. After incubation at 37°C for 1 hour, the pH was neutralized immediately by tittering the infectious virus on MDCK cells as outlined above [50]. Regression analysis of dose response curves

was used to determine which pH caused a 50% reduction in infectious titer ($EC_{50}$). Each experiment was performed in triplicate in three independent biological replicates.

## Stability of influenza virus in stationary droplet

All stability experiments were performed inside a desiccator chamber containing saturated solutions of potassium acetate, magnesium nitrate, or sodium chloride to produce the relative humidities (RH) of 23%, 55%, and 75%. Chambers were maintained in a biosafety cabinet for the duration of the experiment and a Onset HOBO UX100011 data logger was used to collect RH and temperature data. ASL collected from at least twelve HBE patient transwells was pooled and stability experiments were performed three different patient ASL. Virus was mixed with ASL at a ratio of 1:10 and ten 1 µl droplets were placed into a well of a 6-well dish with tissue culture treated plastic. This experiment was done in triplicate. Plates were placed in the desiccator and incubated for 2 hours at the given RH. After 2 hours, virus:ASL droplets were resuspended in 500 µL of Leibovitz's L-15 medium, and titered on MDCK cells by $TCID_{50}$ endpoint method [50]. Decay was determined by subtracting the titer of the virus aged for 2 hours from the titer of the virus that had been deposited and immediately recovered.

## Enzyme-linked lectin assay

Fetuin was diluted to 25 µg/ml in coating buffer (SeraCare) and used to coat a 96-well ultra-high binding polystyrene plate at 4°C overnight. The next day, unbound fetuin was removed by washing plates three times with wash buffer (0.01 M PBS, pH 7.4, 0.05% Tween 20). In a separate 96-well plate, two-fold serial dilutions of $10^{7.5}$ $TCID_{50}$/ml virus stock or 62.5 mU/ml *Clostridium perfringes* neuraminidase (Sigma Aldrich) were performed and then transferred to the fetuin coated plates in duplicate. *Clostridium perfringes* neuraminidase was using to standardize each of the three replicates of the experiment. After a 16-hour incubation at 37°C, the plate was washed 6 times with wash buffer and 100 µL of peroxidase-labeled peanut agglutinin (Sigma Aldrich) was added to each well for 2 hours at room temperature in the dark. Plates were washed three times with wash buffer and 100 µL of O-phenylenediamine dihydrochloride substrate (Sigma Aldrich) was added for 10 minutes and the reaction stopped using an equal volume of 1N sulfuric acid. Absorbance was read at 490nm.

## Ferret screening

Prior to purchase from Triple F Farms (Sayre, PA), sera from three- to five-month-old male ferrets were screened for antibodies against influenza A and B viruses using hemagglutination inhibition (HAI). Briefly, RDE-treated sera were serially diluted two-fold and incubated with eight hemagglutinating units of the following antigens obtained through the International Reagent Resource, Influenza Division, WHO Collaborating Center for Surveillance, Epidemiology and Control of Influenza, Centers for Disease Control and Prevention, Atlanta, GA, USA: 2018–2019 WHO Antigen, Influenza A (H3) Control Antigen (A/Singapore/INFIMH-16–0019/2016), BPL-Inactivated, FR-1606; 2014–2015 WHO Antigen, Influenza A (H1N1)pdm09 Control Antigen (A/California/07/2009 NYMC X-179A), BPL-Inactivated, FR-1184; 2018–2019 WHO Antigen, Influenza B Control Antigen, Victoria Lineage (B/Colorado/06/2017), BPL-Inactivated, FR-1607; 2015–2016 WHO Antigen, Influenza B Control Antigen, Yamagata Lineage (B/Phuket/3073/2013), BPL-Inactivated, FR-1403. After incubation for 15 minutes, an equal volume of 0.5% turkey red blood cells (Lampire Biological Laboratories) were added and incubated for 30 minutes. The reciprocal of the highest dilution of serum that inhibited hemagglutination was determined to be the HAI titer.

## Transmission study design

One hundred and twelve days prior to the start of the transmission studies, six naïve ferrets were inoculated intranasally with $10^6$ $TCID_{50}$ of A/Perth/16/2009 H3N2 in 500 µL of Leibovitz's L-15 medium. These immune imprinted ferrets recovered from primary infection for ~3months (referred to as H3N2-imm recipients) and were used in a direct contact

or airborne transmission study. Four naïve donor ferrets were intranasally inoculated with $10^6$ TCID$_{50}$ of A/Guangdong/ZS023SF005/2023 in 500 µL of Leibovitz's L-15 medium. Twenty-four hours post-donor infection, the H3N2-imm recipient was exposed to the donor for two consecutive days and afterward individually housed for the remainder of the study. For the airborne transmission setup, donor and recipient ferrets are separated by a one-inch-thick perforated divider with directional airflow from donor to recipient, whereas the divider was removed for the direct contact transmission setup. Body weight and clinical signs were recorded upon collection of nasal wash samples on the indicated days.

### Ferret pathogenesis study design

H3N2-imm ferrets were inoculated intranasally with $10^6$ TCID$_{50}$ of A/Perth/16/2009 H3N2 in 500 µL of Leibovitz's L-15 medium one hundred and twelve days prior to the start of pathogenesis studies. Naïve and H3N2-imm ferrets were anesthetized with isoflurane and intranasally infected with $10^6$ TCID$_{50}$ of A/Guangdong/ZS023SF005/2023 in 500 µL of Leibovitz's L-15 medium. At day 3 and 5 post-infection, groups of 3 naïve or H3N2-imm ferrets were euthanized and the following respiratory tissues were collected: pieces of right middle and left cranial lung lobes, trachea, soft palate and nasal turbinates. Tissue samples were weighed, and Leibovitz's L-15 medium was added to make a 10% (lungs) or 5% (trachea) w/v homogenate. The soft palate and nasal turbinates were homogenized in 1 mL of Leibovitz's L-15 medium. Tissues were dissociated using an OMNI GLH homogenizer (OMNI International) and cell debris was removed by centrifugation at 900 xg for 10 minutes. Influenza virus titers were determined by endpoint TCID$_{50}$ assay [50]. The lungs were fixed in 10% neutral buffered formalin and subsequently processed in alcohols for dehydration and embedded in paraffin wax. Embedded sections were cut at 5 µm and stained with hematoxylin and eosin (H&E). The sections were examined 'blind' to experimental groups to eliminate observer bias by a board-certified animal pathologist (LHR).

### Supporting information

**S1 Fig. Lung Pathology of H3N8 infected ferrets.** Representative lung images from A/GD/F005/23 H3N8 infected ferrets at day 5 post-infection. A, B are from ferrets without pre-existing immunity and C, D are from ferrets with H3N2 pre-existing immunity. Scale bar 200 µm.
(PDF)

**S1 Table. A pairwise comparison of the three human H3N8 samples show they differ by multiple nonsynonymous amino acid (AA) substitutions across their genomes.** The table compares the amino acid differences between each pair of isolates, where each row represents one human isolate. Each cell represents the pairwise comparison between that isolate and the isolate in the corresponding column. The identities of each amino acid represent the amino acid encoded for the isolate in each row. For example, A/Henan differs from A/Guangdong at 2 amino acid sites in NP: 50 and 55. A/Henan encodes a C at 50 and an F at 55, while A/Guangdong encodes an F at 50 and an I at 55.
(DOCX)

**S2 Table. Accession numbers for H3N8 strains analyzed in Fig 1.**
(PDF)

### Acknowledgments

We thank Dr. Rachel Duron for critical review and feedback as well as Bailee Werner for her technical contributions.

### Author contributions

**Conceptualization:** Valerie Le Sage, Seema Lakdawala.

**Data curation:** Valerie Le Sage, Maria A. Maltepes, Louise H. Moncla, Seema Lakdawala.

**Formal analysis:** Valerie Le Sage, Maria A. Maltepes, Lora H. Rigatti, Louise H. Moncla, Seema Lakdawala.

**Funding acquisition:** Seema Lakdawala.

**Investigation:** Valerie Le Sage, Michelle N. Vu, Maria A. Maltepes, Shengyang Wang, Brooke A. Snow, Grace A. Merrbach, Alexandra J. Benton, Kylie E. Zirckel, Sarah E. Petnuch, Carly N. Marble, Lora H. Rigatti.

**Methodology:** Valerie Le Sage, Seema Lakdawala.

**Project administration:** Valerie Le Sage, Seema Lakdawala.

**Resources:** James C. Paulson, Elizabeth M. Drapeau, Anita K. McElroy, Scott E. Hensley.

**Supervision:** Valerie Le Sage, James C. Paulson, Louise H. Moncla, Seema Lakdawala.

**Validation:** Valerie Le Sage, Seema Lakdawala.

**Visualization:** Valerie Le Sage, Maria A. Maltepes, Louise H. Moncla, Seema Lakdawala.

**Writing – original draft:** Valerie Le Sage, Seema Lakdawala.

**Writing – review & editing:** Valerie Le Sage, Maria A. Maltepes, Shengyang Wang, Lora H. Rigatti, James C. Paulson, Scott E. Hensley, Louise H. Moncla, Seema Lakdawala.

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
