## [Decision Letter · Decision Letter 0]

11 Nov 2025

Fatal Human H3N8 Influenza Virus has a Moderate Pandemic Risk

PLOS Pathogens

Dear Dr. Lakdawala,

Thank you for submitting your manuscript to PLOS Pathogens. After careful consideration, we feel that it has merit but does not fully meet PLOS Pathogens's publication criteria as it currently stands. Therefore, we invite you to submit a revised version of the manuscript that addresses the points raised during the review process.

We look forward to receiving your revised manuscript.

Kind regards,

Peter Palese

Academic Editor

PLOS Pathogens

Thomas Hoenen

Section Editor

Editor-in-Chief

PLOS Pathogens

Editor-in-Chief

PLOS Pathogens

orcid.org/0000-0002-7699-2064

**Additional Editor Comments :**

Please, carefully address ALL the comments made by the two referees.

**Journal Requirements:**

At this stage, the following Authors/Authors require contributions: Valerie Le Sage, Michelle N. Vu, Maria A. Maltepes, Shengyang Wang, Brooke A. Snow, Grace A. Merrbach, Alexandra J. Benton, Kylie E. Zirckel, Sarah E. Petnuch, Carly N. Marble, Lora H. Rigatti, James C. Paulson, Elizabeth M. Drapeau, Anita K. McElroy, Scott E. Hensley, and Louise H. Moncla. Please ensure that the full contributions of each author are acknowledged in the "Add/Edit/Remove Authors" section of our submission form.

https://journals.plos.org/plospathogens/s/submission-guidelines#loc-parts-of-a-submission

4) We notice that your supplementary Figure, and Table are included in the manuscript file. Please remove them and upload them with the file type 'Supporting Information'. Please ensure that each Supporting Information file has a legend listed in the manuscript after the references list.

5)  Thank you for stating that " full phylogenies from which these were subsetted are publicly viewable and interactive at https://nextstrain.org/groups/moncla-lab/h3nx." This link reaches this message "The dataset "nextstrain.org/groups/moncla-lab/h3nx" doesn't exist." Please amend this to a new link .

7) Please provide a completed 'Competing Interests' statement, including any COIs declared by your co-authors. If you have no competing interests to declare, please state "The authors have declared that no competing interests exist". Otherwise please declare all competing interests beginning with the statement "I have read the journal's policy and the authors of this manuscript have the following competing interests:"

**Reviewers' Comments:**

Reviewer's Responses to Questions

**Part I - Summary**

Reviewer #1: This manuscript by Le Sage and colleagues evaluates the pandemic potential of a low path again H3N8 strain which has made several dead end jumps into humans, causing disease and death. They run through a battery of assays which demonstrate that this virus may be poised for emergence. Overall the manuscript is well written and I believe the results will be of broad interest to the field. More context as to why the authors chose to compare H3N8 to pH1N1 would be helpful. For the antibody history comparison this is ostensibly to show that pH1N1 was able to emerge even in spite of the presence of sera reactivity. It might be intuitive to a lot of readers to make the comparison to seasonal H3 viruses. Additionally, given that there is at least 1 study on H3N8 pandemic potential it would help to highlight what is known and unknown earlier in the paper to put the experiments into better context.

Reviewer #2: The authors reconstituted an avian H3N8 virus using reverse genetics that caused a fatal human infection to investigate its pandemic potentials characterized by replication ability in human bronchial epithelial cells (HBEC), sialic acid receptor binding preference (2,3-linked or 2,6-linked), airborne and direct-contact transmission in pre-immune ferrets and pre-existing antibody responses in humans to it. Despite of lack of cross-neutralizing antibodies in human population, efficient replication of virus in HBEC, dual receptor binding, similar decay rate to pandemic H1N1, the virus was transmitted via direct-contact, but not airborne route in ferrets with immunity to the historical H3N2 virus lower its pandemic potentials to moderate. The study is well conducted with a few comments to be addressed.

**Part II – Major Issues: Key Experiments Required for Acceptance**

Reviewer #1: 1. Understanding if H3N8 can transmit in the face of preexisting H3N2 immunity (which most humans above 2-3yo will have) is critical. However the negative result is difficult to interpret without a positive control. Does this virus transmit to naive animals by aerosol? In the absence of these data the authors should temper their conclusions and acknowledge this limitation.

2. It would be helpful to have a basis of comparison for Fig 3 C and D (even if just historical data). For example are these binding levels similar to seasonal flu viruses? Lower?

Reviewer #2: NA

**Part III – Minor Issues: Editorial and Data Presentation Modifications**

Reviewer #1: 1. Methods state serum were used in fig 1, should this be fig 2?

2. Results section and methods discuss serum from 2020 but these data do not appear to be actually shown in the figure

3. Stats for 4B, D should be shown

4. The methods rescue section mentions MP several times which should be corrected to NP

5. Fig 1 was low resolution and difficult to read.

Reviewer #2: -Please improve the quality of figure 1

-Please add line number and page number

-Figure 4C, please correct the black and red symbols of dots to bars, which more accurately represent the groups.

-Please clarify if the experimental stock of the H3N8 virus was sequence-confirmed?

-Please provide more details on the glycans used in glycan ELISAs, how were they obtained (e.g, from commercial kit or made in-house)

-How did the authors validate the A/Perth/09 H3N2 immunity is waning in ferrets?

-The lack of airborne transmission is found in ferret immunized with H3N2 from 2009. What are the expectations for airborne transmission in naïve individuals or individuals born after 2009? Please discuss.

- In the discussion, shouldn’t the statement “It is important to note that in the 1918 and 2019 H1N1 human pandemics some early isolates exhibited dual receptor specificity resulting from a single mutation from the avian virus progenitor (39-41).” Refer to 2009 H1N1 instead of 2019 H1N1?

-Please described the biosafety levels in which the H3N8 virus was handled since it has a fatal outcome in humans, or justify the use of lower containment conditions

PLOS authors have the option to publish the peer review history of their article (what does this mean? ). If published, this will include your full peer review and any attached files.

**Do you want your identity to be public for this peer review?** For information about this choice, including consent withdrawal, please see our Privacy Policy .

Reviewer #1: No

Reviewer #2: No

**Figure resubmission:**

**Reproducibility:**



---

## [Decision Letter · Decision Letter 1]

19 Feb 2026

Dear Dr. Lakdawala,

We are pleased to inform you that your manuscript 'Fatal Human H3N8 Influenza Virus has a Moderate Pandemic Risk' has been provisionally accepted for publication in PLOS Pathogens.

Best regards,

Peter Palese

Academic Editor

PLOS Pathogens

Thomas Hoenen

Section Editor

PLOS Pathogens

Sumita Bhaduri-McIntosh

Editor-in-Chief

PLOS Pathogens

orcid.org/0000-0003-2946-9497

Michael Malim

Editor-in-Chief

PLOS Pathogens

orcid.org/0000-0002-7699-2064

Reviewer Comments (if any, and for reference):

Reviewer's Responses to Questions

**Part I - Summary**

Reviewer #1: The authors have thoughtfully addressed the concerns of both reviewers and present a significantly strengthened manuscript which will be of interest to the field.

Reviewer #2: No further comments

**Part II – Major Issues: Key Experiments Required for Acceptance**

Reviewer #1: (No Response)

Reviewer #2: (No Response)

**Part III – Minor Issues: Editorial and Data Presentation Modifications**

Reviewer #1: (No Response)

Reviewer #2: (No Response)

PLOS authors have the option to publish the peer review history of their article (what does this mean? ). If published, this will include your full peer review and any attached files.

**Do you want your identity to be public for this peer review?** For information about this choice, including consent withdrawal, please see our Privacy Policy .

Reviewer #1: No

Reviewer #2: No

---

## [Editor Report · Acceptance letter]

Dear Dr. Lakdawala,

We are delighted to inform you that your manuscript, "Fatal Human H3N8 Influenza Virus has a Moderate Pandemic Risk," has been formally accepted for publication in PLOS Pathogens.

Best regards,

Sumita Bhaduri-McIntosh

Editor-in-Chief

PLOS Pathogens

orcid.org/0000-0003-2946-9497

Michael Malim

Editor-in-Chief

PLOS Pathogens

orcid.org/0000-0002-7699-2064